# Association of premature menopause with incident pulmonary hypertension: A cohort study

Michael C. Honigberg[1,2,3,4], Aniruddh P. Patel[1,2,3], Tim Lahm[5,6], Malissa J. Wood[1,4], Jennifer E. Ho[1,3], Puja Kohli[7,8], Pradeep Natarajan[1,2,3]*

1 Cardiology Division, Department of Medicine, Massachusetts General Hospital, Harvard Medical School, Boston, Massachusetts, United States of America, 2 Program in Medical and Population Genetics, Broad Institute of Harvard and MIT, Cambridge, Massachusetts, United States of America, 3 Cardiovascular Research Center and Center for Genomic Medicine, Massachusetts General Hospital, Boston, Massachusetts, United States of America, 4 Cardiology Division, Corrigan Women's Heart Health Program, Massachusetts General Hospital, Boston, Massachusetts, United States of America, 5 Division of Pulmonary, Allergy, Critical Care, and Sleep Medicine, Indiana University School of Medicine, Indianapolis, Indiana, United States of America, 6 Richard L. Roudebush VA Medical Center, Indianapolis, Indiana, United States of America, 7 Vertex Pharmaceuticals, Boston, Massachusetts, United States of America, 8 Division of Pulmonary and Critical Care Medicine, Department of Medicine, Massachusetts General Hospital, Harvard Medical School, Boston, Massachusetts, United States of America

* pnatarajan@mgh.harvard.edu

**Data Availability Statement:** UK Biobank data is available to researchers by application. Researchers may apply for UK Biobank data

## Abstract

### Background

Several forms of pulmonary hypertension (PH) disproportionately affect women. Animal and human studies suggest that estradiol exerts mixed effects on the pulmonary vasculature. Whether premature menopause represents a risk factor for PH is unknown.

### Methods and findings

In this cohort study, women in the UK Biobank aged 40–69 years who were postmenopausal and had complete data available on reproductive history were included. Premature menopause, defined as menopause occurring before age 40 years. Postmenopausal women without premature menopause served as the reference group. The primary outcome was incident PH, ascertained by appearance of a qualifying ICD code in the participant's UK Biobank study record. Of 136,715 postmenopausal women included, 5,201 (3.8%) had premature menopause. Participants were followed up for a median of 11.1 (interquartile range 10.5–11.8) years. The primary outcome occurred in 38 women (0.73%) with premature menopause and 409 (0.31%) without. After adjustment for age, race, ever-smoking, body-mass index, systolic blood pressure, antihypertensive medication use, non-high-density lipoprotein cholesterol, cholesterol-lowering medication use, C-reactive protein, prevalent type 2 diabetes, obstructive sleep apnea, heart failure, mitral regurgitation, aortic stenosis, venous thromboembolism, forced vital capacity (FVC), the forced expiratory volume in 1 second-to-FVC ratio, use of menopausal hormone therapy, and hysterectomy status, premature menopause was independently associated with PH (hazard ratio 2.13, 95% CI 1.31–

access at http://ukbiobank.ac.uk/enable-your-research/apply-for-access.

**Funding:** Dr. Honigberg is supported by the U.S. National Heart, Lung, and Blood Institute [T32HL094301-07]. Dr. Patel is supported by the U.S. National Heart, Lung, and Blood Institute [T32HL007208]. Dr. Lahm is supported by grants from the U.S. National Heart, Lung, and Blood Institute [R01HL144727-01A1] and the U.S. Department of Veterans Affairs [VA Merit Award 1I01BX002042-07]. Dr. Ho is supported by the U. S. National Heart, Lung, and Blood Institute [R01HL134893, R01HL140224]. Dr. Kohli reports employment by Vertex Pharmaceuticals, unrelated to this work. Dr. Natarajan is supported by grants from the U.S. National Heart, Lung, and Blood Institute [R01HL1427, R01HL148565, and R01HL148050], Fondation Leducq [TNE-18CVD04], and a Hassenfeld award from the Massachusetts General Hospital. The authors received no specific funding for this work.

**Competing interests:** Dr. Lahm reports consulting income and speaker fees from Bayer, all unrelated to this work. Dr. Ho reports research support from Bayer, research grant funding from Gilead Sciences, and research supplies from EcoNugenics, all unrelated to this work. Dr. Kohli reports employment by Vertex Pharmaceuticals, unrelated to this work. Dr. Natarajan reports grant support from Amgen, Apple, and Boston Scientific, consulting income from Apple, and spousal employment by Vertex Pharmaceuticals, all unrelated to this work. This does not alter our adherence to PLOS ONE policies on sharing data and materials.

3.23, P<0.001). In analyses of alternate menopausal age thresholds, risk of PH appeared to increase progressively with younger age at menopause ($P_{trend}$ <0.001), with 4.8-fold risk in women with menopause before age 30 years (95% CI 1.82–12.74, P = 0.002). Use of menopausal hormone therapy did not modify the association of premature menopause with PH.

## Conclusions

Premature menopause may represent an independent risk factor for PH in women. Further investigation of the role of sex hormones in PH is needed in animal and human studies to elucidate pathobiology and identify novel therapeutic targets.

## Introduction

Pulmonary hypertension (PH) is a syndrome characterized by elevated mean pulmonary arterial pressure ($\geq$20 mmHg) which may occur in isolation or in association with a range of cardiopulmonary, thrombotic, autoimmune, and infectious diseases [1, 2]. Several forms of PH disproportionately affect women, principally pulmonary arterial hypertension (PAH) [3]. Prior studies suggest a female predominance in PH of up to 4-fold, with greatest sex imbalance observed in PAH and among younger patients [3]. Animal and human studies suggest that sex hormones influence PH development, although observed effects of estrogen and its metabolites have been conflicting across and within animal models [3–5]. Consequently, the role of hormonal modulation in preventing or treating PH and PAH remains uncertain.

The menopausal transition in women is characterized by a marked decline in levels of estrogen and other sex steroid hormones and is associated with accelerated cardiovascular disease risk. Premature menopause, defined by recent guidelines as occurring before age 40 years [6], is associated with elevated risk of diverse cardiovascular conditions independent of conventional risk factors [7]. However, data on the relationship of menopause to PH are limited [8], and whether premature menopause represents a risk factor for PH is unknown. Here, we tested the association of premature menopause with incident PH among postmenopausal women in the UK Biobank. We hypothesized that premature menopause would be independently associated with increased risk of PH.

## Materials and methods

The UK Biobank is a population-based cohort that includes >500,000 adult residents of the United Kingdom who were recruited between 2006–2010. Baseline assessment included medical and reproductive history, vital signs, phlebotomy, and spirometry to obtain forced expiratory volume in 1 second ($FEV_1$) and forced vital capacity (FVC) (Pneumotrac 6800, Vitalograph). Follow-up in the UK Biobank is performed on an ongoing basis through study visits and linkage to national health records and the death register, as described previously [7]. In the present analysis, follow-up occurred through March 2020 for inpatient diagnosis codes and May 2020 for the death register. The UK Biobank is approved by the North West Multi-Centre Research Ethics Committee, and this research was conducted under UK Biobank application #7089. UK Biobank participants provided signed informed consent, and the Massachusetts General Hospital institutional review board approved these analyses.

In this cohort study, all postmenopausal women aged 40–69 years old with complete reproductive data were considered for inclusion. Women with baseline PH, congenital heart disease,

or extreme pulmonary function indices (Z>5 or Z<-5) were excluded. The primary exposure was premature menopause (before age 40 years), ascertained by participant self-report at study enrollment. The study outcome was incident PH, ascertained by the appearance of a qualifying ICD code (ICD-9 4160; ICD-10 I27.0, I27.2) in the study record. Cox proportional hazard models tested the association of premature menopause with incident PH, with adjustment for age, race/ethnicity, ever-smoking, body-mass index, systolic blood pressure, non-high-density lipoprotein cholesterol, antihypertensive and cholesterol-lowering medication, log-transformed C-reactive protein, type 2 diabetes mellitus, obstructive sleep apnea (OSA), heart failure, aortic stenosis, mitral regurgitation, venous thromboembolism, FVC, $FEV_1$-to-FVC ratio, ever-use of menopausal hormone therapy (MHT), and hysterectomy status. Because approximately 95% of the cohort was White, race/ethnicity was categorized as White vs. non-White in analyses. In primary models, subjects with missing data were excluded. Multiple sensitivity analyses were performed, including analyses with imputed missing covariates using the predict() function in R based on age, race/ethnicity, and premature menopause status; analyses excluding women with prevalent cancer and history of hysterectomy; analyses excluding women with prevalent heart failure, aortic stenosis, mitral regurgitation, venous thromboembolism, OSA, and/or COPD; and analyses requiring ≥2 instances of ICD coding to classify incident PH. Missing data are summarized in S1 Table.

Subjects were followed until a diagnosis of PH or until their last clinical encounter or study visit. The proportional hazard assumption was confirmed using Schoenfeld residuals. Two-sided P <0.05 was considered statistically significant. Analyses were performed in R 3.6.0.

## Results

Among 136,715 women in the study cohort (Fig 1), 5,201 (3.8%) experienced premature menopause (mean [SD] age at enrollment 59.9 [5.4] years). Women with premature menopause were more likely to be current or former smokers; had higher BMI; had greater prevalence of atherosclerotic risk factors and coronary artery disease (CAD), chronic obstructive pulmonary disease (COPD), and venous thromboembolism; and had higher C-reactive protein (Table 1).

Follow-up occurred over a median 11.1 (interquartile range 10.5–11.8) years of follow-up. Incident PH was diagnosed in 447 women (overall cumulative incidence 0.33%), including 38 (0.73%) with premature menopause and 409 (0.31%) without, over 57,910 person-years and 1,462,840 person-years of follow-up, respectively. Incidence rates were 6.6/10,000 (95% CI 4.5–8.6/10,000) person-years among women with premature menopause and 2.8/10,000 (95% CI 2.5–3.1/10,000) person-years among women without (difference +3.8/10,000 [95% CI 1.7–5.9/10,000] person-years, P<0.001). Among women with premature menopause, only three incident PH diagnoses occurred in women with surgical premature menopause (n = 629), defined as bilateral oophorectomy before age 40 years, thus further analyses were not stratified by premature menopause subtype. Mean (SD) menopausal age was 48.5 (6.2) years in women with incident PH vs. 49.8 (5.0) years in those without PH (P<0.001). Among the 38 women with premature menopause who developed PH, 9 women (23.7%) had an interim CAD diagnosis and 17 women (44.7%) had an interim heart failure diagnosis between study enrollment and PH diagnosis.

After adjustment only for age, premature menopause was associated with a hazard ratio (HR) of 2.51 for incident PH (95% CI 1.80–3.50, P<0.001). After multivariable adjustment, premature menopause was independently associated with PH (HR 2.13, 95% CI 1.31–3.23, P<0.001) (Fig 2). In exploratory analyses, hazards were similar for PAH (HR 2.06, 95% CI 0.98–4.35, P = 0.058) and secondary PH (HR 2.17, 95% CI 1.32–3.57, P = 0.002). Associations with PH were similar in sensitivity analyses (1) using imputed data to replace missing

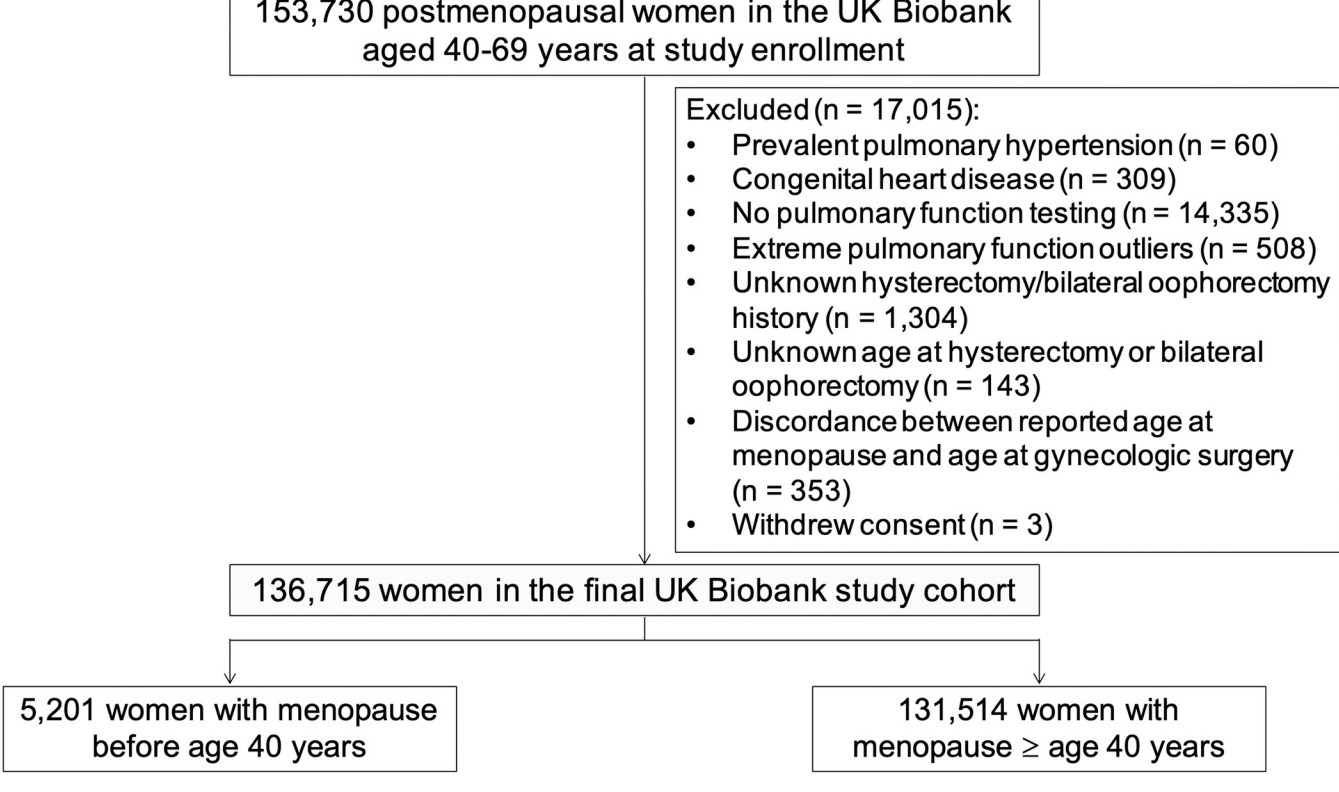

**Fig 1. Creation of the study cohort.**

covariates (HR 2.15, 95% CI 1.50–3.10, P<0.001); (2) excluding 14,441 women with history of cancer or unknown cancer history (HR 1.84, 95% CI 1.13–2.79, P = 0.01); (3) excluding 13,699 women with history of hysterectomy (HR 1.81, 95% CI 0.99–3.32, P = 0.05); (4) excluding 6,769 women with prevalent heart failure, aortic stenosis, mitral regurgitation, venous thromboembolism, OSA, and/or COPD (HR 2.52, 95% CI 1.59–4.00, P<0.001); and (5) requiring ≥2 instances of ICD coding to classify incident PH (HR 2.65, 95% CI 1.38–5.07, P = 0.003).

In analyses of alternate menopausal age thresholds, risk of PH appeared to increase progressively with younger age at menopause ($P_{trend}$ <0.001) (Fig 3). Compared with women who experienced menopause after age 50 years, the HR associated with menopause before age 30 years was 4.82 (95% CI 1.82–12.74, P = 0.002).

Ever-use of MHT was not associated with incident PH (HR 0.90, 95% CI 0.73–1.12, P = 0.36) in multivariable models and did not modify the association of premature menopause with PH ($P_{interaction}$ = 0.63). In addition, there was no significant association of MHT duration with PH (HR 1.01 per year of use, 95% CI 0.99–1.03, P = 0.35). Incorporation of total testosterone, SHBG, and the testosterone-to-SHBG ratio (i.e., free androgen index, a surrogate for free testosterone) in multivariable models did not confound the association of premature menopause with PH.

## Discussion

In a large cohort of postmenopausal women, menopause before age 40 years was independently associated with 2-fold risk of PH. Further increase in PH risk was observed with progressively earlier age at menopause. Higher prevalence of heart failure, valvular heart disease,

**Table 1. Baseline characteristics of the study cohort.**

| Characteristic | Menopause < age 40 y (N = 5,201) | Menopause ≥ age 40 y (N = 131,514) | P-value |
|---|---|---|---|
| Age | 58.5 (7.2) | 60.0 (5.3) | <0.001 |
| Race/ethnicity | | | <0.001 |
| • White | 4,923 (94.7%) | 125,834 (95.7%) | |
| • Black | 98 (1.9%) | 1,451 (1.1%) | |
| • Asian | 90 (1.7%) | 2,276 (1.7%) | |
| • Mixed | 37 (0.7%) | 606 (0.5%) | |
| • Other | 53 (1.0%) | 1,347 (1.0%) | |
| Current or former smoking | 2,546 (49.0%) | 54,127 (41.1%) | <0.001 |
| Parity, median [IQR] | 2 [1, 3] | 2 [1, 3] | 0.03 |
| History of hypertensive disorder of pregnancy | 41 (0.8%) | 889 (0.7%) | 0.37 |
| Age at menopause | 35.2 (4.0) | 50.3 (4.1) | <0.001 |
| History of hysterectomy (before or after menopause) | 2,864 (55.1%) | 10,835 (8.2%) | <0.001 |
| History of bilateral oophorectomy (before or after menopause) | 849 (16.3%) | 6,210 (4.7%) | <0.001 |
| History of any cancer | 798 (15.4%) | 13,244 (10.1%) | <0.001 |
| Ever-use of menopausal hormone therapy | 3,801 (73.3%) | 57,462 (43.8%) | <0.001 |
| Body-mass index | 28.0 (5.5) | 27.0 (4.9) | <0.001 |
| Systolic blood pressure, mmHg | 138.9 (20.4) | 140.3 (20.2) | <0.001 |
| Diastolic blood pressure, mmHg | 80.9 (10.5) | 81.0 (10.4) | 0.38 |
| Hypertension | 1,679 (32.3%) | 36,075 (27.4%) | <0.001 |
| Hyperlipidemia | 925 (17.8%) | 16,702 (12.7%) | <0.001 |
| Type 2 diabetes mellitus | 147 (2.8%) | 2,009 (1.5%) | <0.001 |
| Coronary artery disease | 120 (2.3%) | 1,463 (1.1%) | <0.001 |
| Obstructive sleep apnea | 26 (0.5%) | 378 (0.3%) | 0.008 |
| Heart failure | 26 (0.5%) | 300 (0.2%) | <0.001 |
| Aortic stenosis | 10 (0.2%) | 107 (0.1%) | 0.01 |
| Mitral regurgitation | 13 (0.2%) | 191 (0.1%) | 0.08 |
| Venous thromboembolism | 261 (5.0%) | 3,516 (2.7%) | <0.001 |
| Chronic obstructive pulmonary disease | 183 (3.6%) | 2,082 (1.6%) | <0.001 |
| Antihypertensive medication use | 1,247 (24.0%) | 26,305 (20.0%) | <0.001 |
| Cholesterol-lowering medication use | 1,126 (21.6%) | 19,354 (14.7%) | <0.001 |
| Low-density lipoprotein cholesterol, mg/dL | 141.9 (34.5) | 145.4 (33.7) | <0.001 |
| Non-high-density lipoprotein cholesterol, mg/dL | 168.9 (43.1) | 171.8 (41.5) | <0.001 |
| C-reactive protein, mg/L, median [IQR] | 1.9 (0.9–3.9) | 1.4 (0.7–2.9) | <0.001 |
| Total testosterone, nmol/L, median [IQR] | 0.95 [0.67, 1.31] | 0.98 [0.70, 1.33] | <0.001 |
| Sex hormone-binding globulin, nmol/L, median [IQR] | 54.3 [38.2, 76.9] | 55.0 [39.5, 74.4] | 0.78 |
| Ratio of total testosterone to sex hormone-binding globulin, median [IQR] | 0.02 [0.01, 0.03] | 0.02 [0.01, 0.03] | 0.17 |
| Forced expiratory volume in 1 second, L | 2.2 (0.5) | 2.3 (0.5) | <0.001 |
| Forced vital capacity, L | 2.9 (0.6) | 3.0 (0.6) | <0.001 |
| Ratio of forced expiratory volume in 1 second to forced vital capacity | 0.8 (0.1) | 0.8 (0.1) | 0.05 |

Values are displayed as mean (standard deviation) for continuous variables and N (%) for categorical variables unless otherwise specified.

OSA, COPD, and venous thromboembolism among women with premature menopause did not explain the observed increased risk for PH.

While estrogen and its metabolites can worsen adverse pulmonary vascular remodeling in established PAH, estrogens also exert a variety of vasculoprotective effects that may be beneficial in PAH and other forms of PH [3]. In women with systemic sclerosis, PAH typically does

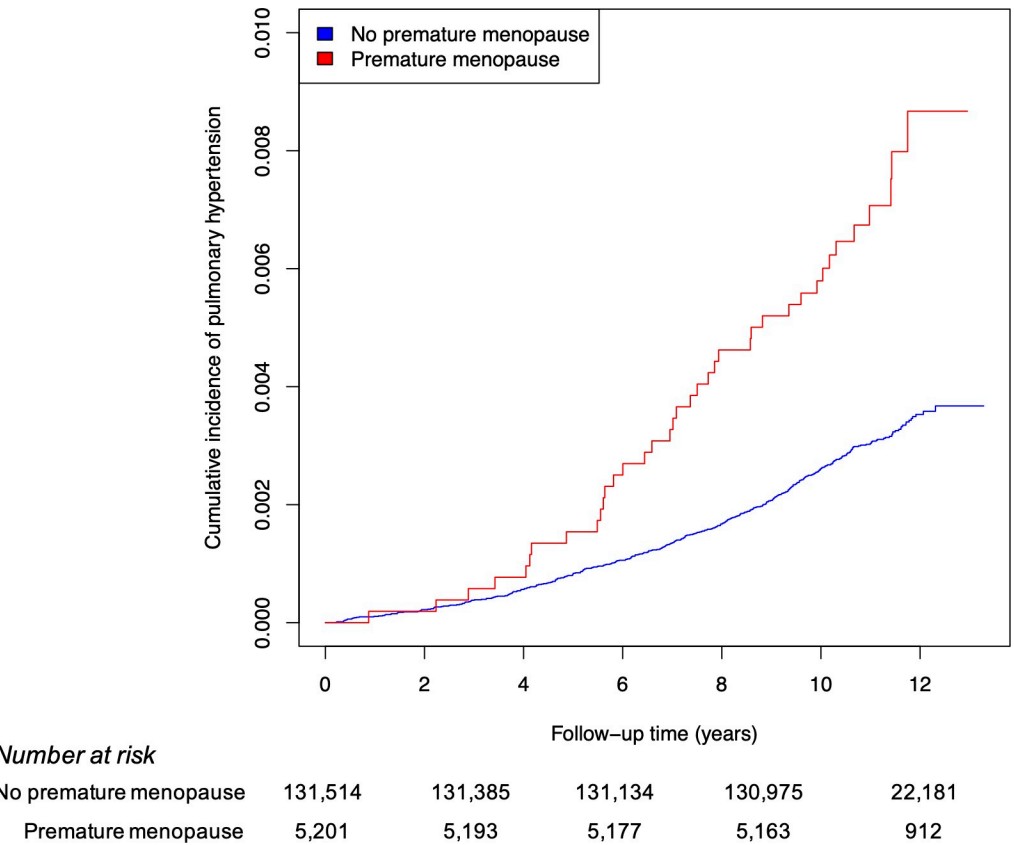

**Fig 2. Cumulative incidence of pulmonary hypertension by premature menopause status.** Premature menopause was independently associated with an increased risk of incident pulmonary hypertension among postmenopausal women in the UK Biobank, with an adjusted hazard ratio of 2.13 (95% confidence interval 1.31–3.23, P<0.001). The hazard ratio was adjusted for age, race/ethnicity, ever-smoking, body-mass index, systolic blood pressure, non-high-density lipoprotein cholesterol, antihypertensive and cholesterol-lowering medication, C-reactive protein, type 2 diabetes mellitus, obstructive sleep apnea, heart failure, aortic stenosis, mitral regurgitation, venous thromboembolism, FVC, $FEV_1$-to-FVC ratio, use of menopausal hormone therapy, and hysterectomy status.

not arise until after menopause, suggesting protective estrogen effects [8]. Female PAH patients exhibit more favorable hemodynamics than male patients, but these differences disappear in women >45 years of age [9]. Furthermore, in a small randomized trial of individuals with PAH receiving the aromatase inhibitor anastrozole, changes in right ventricular function and natriuretic peptides were heterogeneous among individuals [10]. A small trial of fulvestrant, an estrogen receptor antagonist, did not demonstrate benefit in PAH [11]. Notably, hormone therapy use was not associated with PH in our cohort. Our findings underscore the complex role of sex hormones in PH/PAH and indicate that estrogen signaling is not a "black-and-white" phenomenon.

This study has limitations. Incident PH diagnoses were ascertained from ICD codes rather than invasive hemodynamics; primary analyses therefore tested associations with overall PH rather than PH subtypes (e.g., PAH vs. secondary PH). Data on MHT doses and preparations were unavailable. "Healthy participant bias" in the UK Biobank [12] may bias the magnitude of estimated associations toward the null. More than 95% of the study sample was White, and whether results generalize to diverse populations requires further study. Finally, we were unable to assess the role of estradiol as levels were unavailable in 95% of the cohort (mainly due to levels falling below the assay's reportable limit in the UK Biobank).

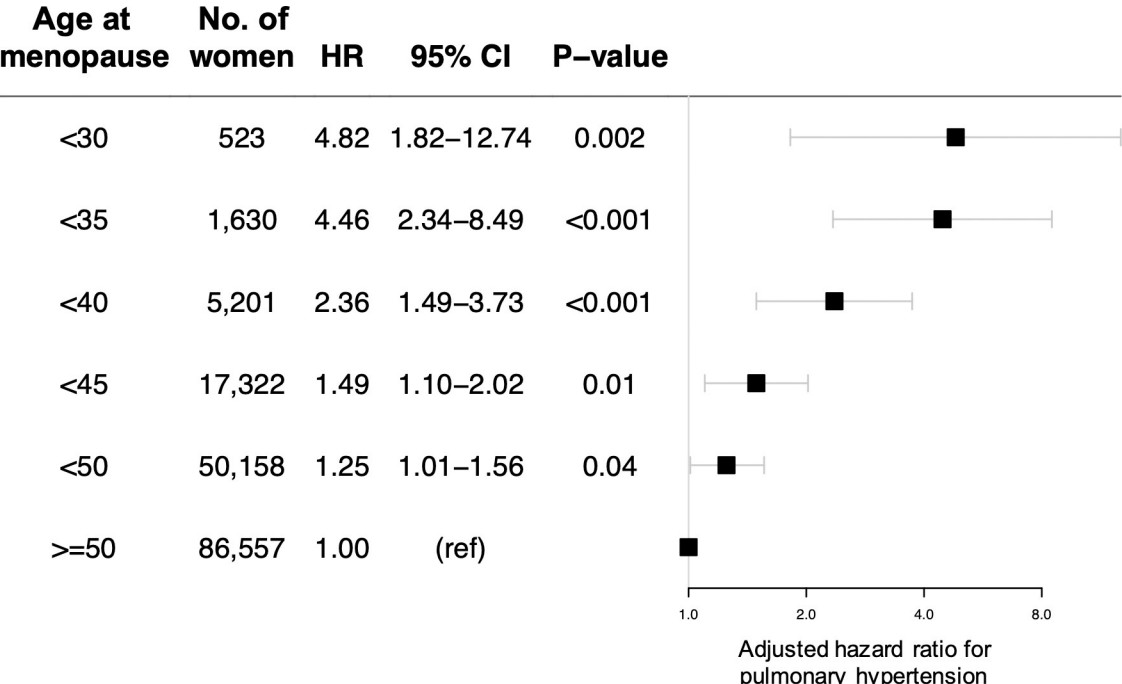

| Age at menopause | No. of women | HR | 95% CI | P–value |
|---|---|---|---|---|
| <30 | 523 | 4.82 | 1.82–12.74 | 0.002 |
| <35 | 1,630 | 4.46 | 2.34–8.49 | <0.001 |
| <40 | 5,201 | 2.36 | 1.49–3.73 | <0.001 |
| <45 | 17,322 | 1.49 | 1.10–2.02 | 0.01 |
| <50 | 50,158 | 1.25 | 1.01–1.56 | 0.04 |
| >=50 | 86,557 | 1.00 | (ref) | |

**Fig 3. Hazard ratios (95% confidence intervals) for incident pulmonary hypertension associated with different menopausal age thresholds.** Groups are inclusive of all women with age at menopause below the listed cutoff. The reference group for all models is menopause at age 50 years or older. Models are adjusted for age, race/ethnicity, ever-smoking, body-mass index, systolic blood pressure, antihypertensive medication use, non-high-density lipoprotein cholesterol, cholesterol-lowering medication use, C-reactive protein, prevalent type 2 diabetes mellitus, obstructive sleep apnea, heart failure, aortic stenosis, mitral regurgitation, and venous thromboembolism, forced vital capacity, the ratio of the forced expiratory volume in 1 second to forced vital capacity, ever-use of menopausal hormone therapy, and hysterectomy status. P-values are derived from multivariable Cox proportional hazard models.

Premature menopause may represent an independent risk factor for PH. Further investigation of the role of sex hormones in PH is needed to elucidate pathobiology and identify novel therapeutic targets.

## Supporting information

**S1 Checklist.**
(DOCX)

**S1 Table. Summary of data missingness in the study cohort.**
(DOCX)

## Acknowledgments

This work was conducted under UK Biobank application #7089. Researchers may apply for UK Biobank data access (https://www.ukbiobank.ac.uk/).

## Author Contributions

**Conceptualization:** Michael C. Honigberg, Tim Lahm, Pradeep Natarajan.

**Data curation:** Michael C. Honigberg, Aniruddh P. Patel.

**Formal analysis:** Michael C. Honigberg, Aniruddh P. Patel, Malissa J. Wood, Jennifer E. Ho, Pradeep Natarajan.

**Funding acquisition:** Pradeep Natarajan.

**Investigation:** Michael C. Honigberg, Tim Lahm.

**Methodology:** Michael C. Honigberg, Tim Lahm, Pradeep Natarajan.

**Supervision:** Tim Lahm, Malissa J. Wood, Jennifer E. Ho, Puja Kohli, Pradeep Natarajan.

**Writing – original draft:** Michael C. Honigberg.

**Writing – review & editing:** Aniruddh P. Patel, Tim Lahm, Malissa J. Wood, Jennifer E. Ho, Puja Kohli, Pradeep Natarajan.

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
