## [Decision Letter · Decision Letter 0]

18 Jan 2021

PONE-D-20-35592

Association of premature menopause with incident pulmonary hypertension: A cohort study

PLOS ONE

Dear Dr. Natarajan,

Thank you for submitting your manuscript to PLOS ONE. After careful consideration, we feel that it has merit but does not fully meet PLOS ONE’s publication criteria as it currently stands. Therefore, we invite you to submit a revised version of the manuscript that addresses the points raised during the review process.

We look forward to receiving your revised manuscript.

Kind regards,

Yoshihiro Fukumoto

Academic Editor

PLOS ONE

Journal Requirements:

2.) During our internal checks, the in-house editorial staff noted that you conducted research or obtained samples in another country. Please check the relevant national regulations and laws applying to foreign researchers and state whether you obtained the required permits and approvals. Please address this in your ethics statement in both the manuscript and submission information.

3.) Thank you for stating the following in the Funding Section of your manuscript:

'Dr. Honigberg is supported by the U.S. National Heart, Lung, and Blood Institute

[T32HL094301-07]. Dr. Patel is supported by the U.S. National Heart, Lung, and Blood Institute

[T32HL007208]. Dr. Lahm is supported by grants from the U.S. National Heart, Lung, and

Blood Institute [R01HL144727-01A1] and the U.S. Department of Veterans Affairs [VA Merit

Award 1I01BX002042-07]. Dr. Ho is supported by the U.S. National Heart, Lung, and Blood

Institute [R01HL134893, R01HL140224]. Dr. Natarajan is supported by grants from the U.S.

National Heart, Lung, and Blood Institute [R01HL1427, R01HL148565, and R01HL148050],

Fondation Leducq [TNE-18CVD04], and a Hassenfeld award from the Massachusetts General

Hospital.'

We note that you have provided funding information that is not currently declared in your Funding Statement. However, funding information should not appear in the Funding section or other areas of your manuscript. We will only publish funding information present in the Funding Statement section of the online submission form.

'The authors received no specific funding for this work.'

4.) Thank you for stating the following in the Competing Interests section:

'Dr. Lahm reports consulting income and speaker fees from Bayer, all unrelated to this work. Dr. Ho reports research support from Bayer, research grant funding from Gilead Sciences, and research supplies from EcoNugenics, all unrelated to this work. Dr. Kohli reports employment by Vertex Pharmaceuticals. Dr. Natarajan reports grant support from Amgen, Apple, and Boston Scientific, consulting income from Apple, and spousal employment by Vertex Pharmaceuticals, all unrelated to this work.'

We note that one or more of the authors have an affiliation to the commercial funders of this research study : Vertex Pharmaceuticals.

a.) Please provide an amended Funding Statement declaring this commercial affiliation, as well as a statement regarding the Role of Funders in your study. If the funding organization did not play a role in the study design, data collection and analysis, decision to publish, or preparation of the manuscript and only provided financial support in the form of authors' salaries and/or research materials, please review your statements relating to the author contributions, and ensure you have specifically and accurately indicated the role(s) that these authors had in your study. You can update author roles in the Author Contributions section of the online submission form.

b.) Please also provide an updated Competing Interests Statement declaring this commercial affiliation along with any other relevant declarations relating to employment, consultancy, patents, products in development, or marketed products, etc.  

5.) We note that you have indicated that data from this study are available upon request. PLOS only allows data to be available upon request if there are legal or ethical restrictions on sharing data publicly. For information on unacceptable data access restrictions, please see http://journals.plos.org/plosone/s/data-availability#loc-unacceptable-data-access-restrictions.

6.) Please include captions for your Supporting Information files at the end of your manuscript, and update any in-text citations to match accordingly. Please see our Supporting Information guidelines for more information: http://journals.plos.org/plosone/s/supporting-information.

Reviewers' comments:

Reviewer's Responses to Questions

**Comments to the Author**

1. Is the manuscript technically sound, and do the data support the conclusions?

Reviewer #1: Yes

Reviewer #2: Yes

2. Has the statistical analysis been performed appropriately and rigorously? 

Reviewer #1: Yes

Reviewer #2: Yes

3. Have the authors made all data underlying the findings in their manuscript fully available?

Reviewer #1: Yes

Reviewer #2: Yes

4. Is the manuscript presented in an intelligible fashion and written in standard English?

Reviewer #1: Yes

Reviewer #2: Yes

5. Review Comments to the Author

Reviewer #1: This is an interesting manuscript describing the association of premature menopause with incident PH in UK Biobank participants.

I have one comment.

I recommend the authors to add the data of men for reference.

Reviewer #2: In this manuscript, the authors described the possibility of premature menopause as an independent risk factor for PH in women. This study addresses very important issue and indicated interesting results. I have one question. The authors described that incidence rates of PH were 6.6/10,000 person-years among women with premature menopause and 2.8/10,000 person-years among women without. I think the incident rate of PH in both groups seems to be a little higher. Please give your opinion about this issue.

6. PLOS authors have the option to publish the peer review history of their article (what does this mean?). If published, this will include your full peer review and any attached files.

Reviewer #1: No

Reviewer #2: No

---

## [Author Response · Author response to Decision Letter 0]

20 Jan 2021

Response to Editors and Reviewers

We thank the editors and reviewers for their feedback and for the opportunity to submit a revised manuscript.

Editors

Author Response: We revised the manuscript and supporting documents in accordance with the journal’s style requirements.

Manuscript Change: We revised the manuscript and supporting documents in accordance with the journal’s style requirements.

2.) During our internal checks, the in-house editorial staff noted that you conducted research or obtained samples in another country. Please check the relevant national regulations and laws applying to foreign researchers and state whether you obtained the required permits and approvals. Please address this in your ethics statement in both the manuscript and submission information.

Author Response: We have a valid, approved data agreement with the UK Biobank, which is a resource available to researchers worldwide with appropriate credentials upon application.

Manuscript Change: “The UK Biobank is approved by the North West Multi-Centre Research Ethics Committee, and this research was conducted under UK Biobank application #7089.”

3.) Thank you for stating the following in the Funding Section of your manuscript:

'Dr. Honigberg is supported by the U.S. National Heart, Lung, and Blood Institute

[T32HL094301-07]. Dr. Patel is supported by the U.S. National Heart, Lung, and Blood Institute

[T32HL007208]. Dr. Lahm is supported by grants from the U.S. National Heart, Lung, and

Blood Institute [R01HL144727-01A1] and the U.S. Department of Veterans Affairs [VA Merit

Award 1I01BX002042-07]. Dr. Ho is supported by the U.S. National Heart, Lung, and Blood

Institute [R01HL134893, R01HL140224]. Dr. Natarajan is supported by grants from the U.S.

National Heart, Lung, and Blood Institute [R01HL1427, R01HL148565, and R01HL148050],

Fondation Leducq [TNE-18CVD04], and a Hassenfeld award from the Massachusetts General

Hospital.'

We note that you have provided funding information that is not currently declared in your Funding Statement. However, funding information should not appear in the Funding section or other areas of your manuscript. We will only publish funding information present in the Funding Statement section of the online submission form.

'The authors received no specific funding for this work.'

Author Response: We removed this text from the manuscript file. This paragraph summarizes the authors’ current research support. However, as stated, the authors received no specific funding for this particular work.

Manuscript Change: We updated the funding statement to read: “Dr. Honigberg is supported by the U.S. National Heart, Lung, and Blood Institute [T32HL094301-07]. Dr. Patel is supported by the U.S. National Heart, Lung, and Blood Institute [T32HL007208]. Dr. Lahm is supported by grants from the U.S. National Heart, Lung, and Blood Institute [R01HL144727-01A1] and the U.S. Department of Veterans Affairs [VA Merit Award 1I01BX002042-07]. Dr. Ho is supported by the U.S. National Heart, Lung, and Blood Institute [R01HL134893, R01HL140224]. Dr. Kohli reports employment by Vertex Pharmaceuticals, unrelated to this work. Dr. Natarajan is supported by grants from the U.S. National Heart, Lung, and Blood Institute [R01HL1427, R01HL148565, and R01HL148050], Fondation Leducq [TNE-18CVD04], and a Hassenfeld award from the Massachusetts General Hospital. The authors received no specific funding for this work.”

4.) Thank you for stating the following in the Competing Interests section:

'Dr. Lahm reports consulting income and speaker fees from Bayer, all unrelated to this work. Dr. Ho reports research support from Bayer, research grant funding from Gilead Sciences, and research supplies from EcoNugenics, all unrelated to this work. Dr. Kohli reports employment by Vertex Pharmaceuticals, unrelated to this work. Dr. Natarajan reports grant support from Amgen, Apple, and Boston Scientific, consulting income from Apple, and spousal employment by Vertex Pharmaceuticals, all unrelated to this work.'

We note that one or more of the authors have an affiliation to the commercial funders of this research study : Vertex Pharmaceuticals.

a.) Please provide an amended Funding Statement declaring this commercial affiliation, as well as a statement regarding the Role of Funders in your study. If the funding organization did not play a role in the study design, data collection and analysis, decision to publish, or preparation of the manuscript and only provided financial support in the form of authors' salaries and/or research materials, please review your statements relating to the author contributions, and ensure you have specifically and accurately indicated the role(s) that these authors had in your study. You can update author roles in the Author Contributions section of the online submission form.

Author Response: To clarify, Vertex Pharmaceuticals did not fund this study and had no role in this study. Dr. Kohli’s employment with Vertex is entirely unrelated to this work. There was no commercial funding for this study. As stated, the authors received no specific funding for this work.

Manuscript Change: We updated the funding statement to read: “Dr. Honigberg is supported by the U.S. National Heart, Lung, and Blood Institute [T32HL094301-07]. Dr. Patel is supported by the U.S. National Heart, Lung, and Blood Institute [T32HL007208]. Dr. Lahm is supported by grants from the U.S. National Heart, Lung, and Blood Institute [R01HL144727-01A1] and the U.S. Department of Veterans Affairs [VA Merit Award 1I01BX002042-07]. Dr. Ho is supported by the U.S. National Heart, Lung, and Blood Institute [R01HL134893, R01HL140224]. Dr. Kohli reports employment by Vertex Pharmaceuticals, unrelated to this work. Dr. Natarajan is supported by grants from the U.S. National Heart, Lung, and Blood Institute [R01HL1427, R01HL148565, and R01HL148050], Fondation Leducq [TNE-18CVD04], and a Hassenfeld award from the Massachusetts General Hospital. The authors received no specific funding for this work.”

b.) Please also provide an updated Competing Interests Statement declaring this commercial affiliation along with any other relevant declarations relating to employment, consultancy, patents, products in development, or marketed products, etc. 

Please know it is PLOS ONE policy for corresponding authors to declare, on behalf of all authors, all potential competing interests for the purposes of transparency. PLOS defines a competing interest as anything that interferes with, or could reasonably be perceived as interfering with, the full and objective presentation, peer review, editorial decision-making, or publication of research or non-research articles submitted to one of the journals. Competing interests can be financial or non-financial, professional, or personal. Competing interests can arise in relationship to an organization or another person. Please follow this link to our website for more details on competing interests:http://journals.plos.org/plosone/s/competing-interests

Author Response: We updated the Competing Interests statement. 

Manuscript Change: We updated the Competing Interests statement as follows: “Dr. Lahm reports consulting income and speaker fees from Bayer, all unrelated to this work. Dr. Ho reports research support from Bayer, research grant funding from Gilead Sciences, and research supplies from EcoNugenics, all unrelated to this work. Dr. Kohli reports employment by Vertex Pharmaceuticals, unrelated to this work. Dr. Natarajan reports grant support from Amgen, Apple, and Boston Scientific, consulting income from Apple, and spousal employment by Vertex Pharmaceuticals, all unrelated to this work. This does not alter our adherence to PLOS ONE policies on sharing data and materials.”

5.) We note that you have indicated that data from this study are available upon request. PLOS only allows data to be available upon request if there are legal or ethical restrictions on sharing data publicly. For information on unacceptable data access restrictions, please see http://journals.plos.org/plosone/s/data-availability#loc-unacceptable-data-access-restrictions.

Author Response: As is presumably well known to the PLOS ONE staff, the UK Biobank does not permit us to share its data publicly. However, as stated, UK Biobank data is available to researchers by application. Researchers may apply for UK Biobank data access at http://ukbiobank.ac.uk/enable-your-research/apply-for-access. 

Manuscript Change: We provide an updated web address in the cover letter for researchers interested in obtaining UK Biobank data access.

Author Response: As above, the UK Biobank does not permit us to share its data publicly.

6.) Please include captions for your Supporting Information files at the end of your manuscript, and update any in-text citations to match accordingly. Please see our Supporting Information guidelines for more information:http://journals.plos.org/plosone/s/supporting-information.

Author Response: We updated supporting information and citations thereof in accordance with journal style guidelines.

Reviewer #1 

This is an interesting manuscript describing the association of premature menopause with incident PH in UK Biobank participants.

I have one comment. I recommend the authors to add the data of men for reference.

Author Response: We thank the Reviewer for this suggestion. We added additional background summarizing data on female predominance in pulmonary hypertension.

Manuscript Change: Introduction: “Prior studies suggest a female predominance in PH of up to 4-fold, with greatest sex imbalance observed in PAH and among younger patients [3].”

Reviewer #2

In this manuscript, the authors described the possibility of premature menopause as an independent risk factor for PH in women. This study addresses very important issue and indicated interesting results. I have one question. The authors described that incidence rates of PH were 6.6/10,000 person-years among women with premature menopause and 2.8/10,000 person-years among women without. I think the incident rate of PH in both groups seems to be a little higher. Please give your opinion about this issue.

Author Response: We thank the Reviewer for the opportunity to clarify our results. As stated, the cumulative incidence rates of pulmonary hypertension over a median 11.1 years of follow-up were 0.73% and 0.31% in women with and without a history of premature menopause, respectively. Incident PH was diagnosed in 38 women with premature menopause over 57,910 person-years of follow-up, and in 409 women without premature menopause over 1,462,840 years of follow-up. These numbers correspond to incidence rates of 6.6/10,000 person-years and 2.8/10,000 person-years in women with and without a history of premature menopause, respectively. We added total person-years of follow-up to the results so that readers can more easily understand how these incidence rates were calculated.

Manuscript Change: Results: “Follow-up occurred over a median 11.1 (interquartile range 10.5-11.8) years of follow-up. Incident PH was diagnosed in 447 women (overall cumulative incidence 0.33%), including 38 (0.73%) with premature menopause and 409 (0.31%) without, over 57,910 person-years and 1,462,840 person-years of follow-up, respectively. Incidence rates were 6.6/10,000 (95% CI 4.5-8.6/10,000) person-years among women with premature menopause and 2.8/10,000 (95% CI 2.5-3.1/10,000) person-years among women without (difference +3.8/10,000 [95% CI 1.7-5.9/10,000] person-years, P<0.001).”

---

## [Decision Letter · Decision Letter 1]

8 Feb 2021

Association of premature menopause with incident pulmonary hypertension: A cohort study

PONE-D-20-35592R1

Dear Dr. Natarajan,

We’re pleased to inform you that your manuscript has been judged scientifically suitable for publication and will be formally accepted for publication once it meets all outstanding technical requirements.

Kind regards,

Yoshihiro Fukumoto

Academic Editor

PLOS ONE

Additional Editor Comments (optional):

Reviewers' comments:

Reviewer's Responses to Questions

**Comments to the Author**

1. If the authors have adequately addressed your comments raised in a previous round of review and you feel that this manuscript is now acceptable for publication, you may indicate that here to bypass the “Comments to the Author” section, enter your conflict of interest statement in the “Confidential to Editor” section, and submit your "Accept" recommendation.

Reviewer #1: All comments have been addressed

Reviewer #2: All comments have been addressed

2. Is the manuscript technically sound, and do the data support the conclusions?

Reviewer #1: Yes

Reviewer #2: Yes

3. Has the statistical analysis been performed appropriately and rigorously? 

Reviewer #1: Yes

Reviewer #2: Yes

4. Have the authors made all data underlying the findings in their manuscript fully available?

Reviewer #1: Yes

Reviewer #2: Yes

5. Is the manuscript presented in an intelligible fashion and written in standard English?

Reviewer #1: Yes

Reviewer #2: Yes

6. Review Comments to the Author

Reviewer #1: (No Response)

Reviewer #2: In this manuscript, the authors described the possibility of premature menopause as an independent risk factor for PH in women. This study addresses very important issue and indicated interesting results. The response to my comment is fully addressed in the revised manuscript. I have no more comments.

7. PLOS authors have the option to publish the peer review history of their article (what does this mean?). If published, this will include your full peer review and any attached files.

Reviewer #1: No

Reviewer #2: No

---

## [Editor Report · Acceptance letter]

15 Feb 2021

PONE-D-20-35592R1 

Association of premature menopause with incident pulmonary hypertension: A cohort study 

Dear Dr. Natarajan:

I'm pleased to inform you that your manuscript has been deemed suitable for publication in PLOS ONE. Congratulations! Your manuscript is now with our production department. 

Kind regards, 

on behalf of

Dr. Yoshihiro Fukumoto 

Academic Editor

PLOS ONE